# A Study of PI Controller Tuning Methods Using the Internal Model Control Guide for a Ship Central Cooling System as a Multi-Input, Single-Output System

Tae-Youl Jeon [1] and Byung-Gun Jung [2,*]

1    Pacific & MEA Operations, American Bureau of Shipping, Ulsan 44032, Republic of Korea; tjeon@eagle.org
2    Division of Marine System Engineering, Korea Maritime and Ocean University, 727, Taejong-ro, Yeongdo-gu, Busan 49112, Republic of Korea
*    Correspondence: bgjung@kmou.ac.kr

**Abstract:** Since the variable-speed seawater pump and the three-way valve of a ship's central cooling system are forms of feedback from the same output signal, these controllers may cause interference when they are not coordinated. Therefore, studying an efficient control tuning method for a central cooling water system is necessary. In this study, a central cooling water system is modeled using Matlab Simulink by utilizing an actual operation dataset, and unknown parameters are estimated for fine-tuning. The simulation model is then verified using the second operating dataset. Additionally, transfer functions are developed for the freshwater output temperature against the three-way valve openness input and the electrical power frequency input to the seawater pump motor, supposing that the two systems are independent. Then, the two PI controllers are tuned using internal model control (IMC) filters. Moreover, the modified internal model control tuning method is suggested, using the character of the central cooling system, which has a large time constant for the heat exchanger system with the seawater pump and a small time constant for the three-way valve system. This method simplifies the tuning of the two combined PI controllers, enhancing the seawater pump's overall efficiency and the three-way valve's operation. This study presents the proposed tuning methods based on the IMC filter and modified IMC guide, which confirmed the simple determination of the gain values of the PI controller with the efficient control of the rotational speed of the seawater pump and the three-way valve with reasonable control of the freshwater outlet temperature.

**Keywords:** ship's central cooling system; internal model control (IMC); multi-input single-output (MISO)





## 1. Introduction

The efficient control of a ship's central cooling system, which is applied with variable-speed control of the seawater (S.W.) pump, is essential for maintaining steady temperatures and ensuring the proper function of the onboard equipment [1–4]. This study focused on two control applications within the central cooling system, regulating the three-way valve's openness, and controlling the seawater pump's rotational speed, which was driven by a variable-frequency drive (VFD) motor. Both objectives of the controllers contributed to achieving the desired outlet temperature of the freshwater (F.W.).

Harmonizing the control actions of the two controllers is essential for ensuring the effective and efficient operation of the ship's central cooling system [5–7]. As the three-way valve regulates the F.W. flow into the heat exchanger (HEX) and bypasses the flow, it plays a crucial role in controlling the sudden change in the inlet F.W. temperature. The three-way valve system can effectively manage input temperature fluctuations and maintain a stable cooling performance by adjusting the valve's openness. On the other hand, the S.W. pump controls the flow of S.W. circulation throughout the HEX. It indirectly impacts the

F.W. outlet temperature by adjusting the flow rate, allowing for efficient heat transfer and temperature regulation.

Finding a balance between the control actions of the three-way valve and the S.W. pump is a challenge. If the three-way valve operates too aggressively, constantly changing its position, this can lead to unstable F.W. output temperature control and inefficient energy usage. Conversely, if the S.W. pump operates at a high rotational speed, it may result in excessive energy consumption and unnecessary stress on the equipment. Therefore, it is essential to harmonize the control actions of the two controllers in order to maintain temperature stability with energy efficiency [6,8].

In order to overcome this challenge, two controller tuning have been considered. The first method applied the internal model control (IMC) method, which is well-known for its effectiveness in handling systems with time delays and complex dynamics [9–11]. Fine-tuning the controllers with the IMC method necessitates the decoupling of the two systems and the development of their transfer functions [12]. However, for the first proposed IMC-based PI controller tuning method, it was supposed that the two systems are not affected by each other, enabling one to create simple systems and then develop simple transfer functions. This method could be used to effectively tune the PI controller so as to regulate the three-way valve's openness and S.W. pump's rotational speed through the IMC filters.

The second proposed method involves setting the integral gain to be larger than the proportional gain for the S.W. pump rotational speed PI controller, this being derived from the IMC method. This adjustment, achieved by switching the proportional and integral gains, emphasizes the importance of maintaining a lower rotational speed for the S.W. pump. Increasing the integral gain relative to the proportional gain, the controller emphasizes the elimination of steady-state errors and the control of the pump speed at a lower level, thereby promoting energy efficiency.

Simulation-based evaluations were conducted using a ship's central cooling system model to assess the performance of both tuning methods. The simulations demonstrated the effectiveness of the IMC-based PI controller tuning method in achieving temperature stability and efficiently running through step disturbances in the F.W. and S.W. supply temperatures. Additionally, the proposed adjustments for the S.W. pump rotational speed controller showed that this method resulted in reducing the S.W. pump speed while maintaining the desired cooling performance.

This paper explores a simple way of tuning the three-way valve and the S.W. pump controllers in the ship's central cooling system, which is a multi-input, single-output (MISO) system. The goal was to achieve stable temperature control and a reduced S.W. pump motor rotational speed, which are closely related to energy efficiency. The proposed tuning methods allowed for the coordinated operation of the two controllers, resulting in an improved system performance and reduced S.W. pump speed. The research outcomes contribute to the advancement of ship application control techniques, ensuring the central cooling system's effective and efficient operation.

## 2. Simulation Modeling of the Ship's Central Cooling System

### 2.1. Configuration of Central Cooling System on a Ship

The intended modeling of the central cooling system of the actual ship is shown in Figure 1. It consisted of an S.W. pump that supplied S.W., an F.W. pump that circulated F.W., an HEX that cooled the hot F.W. through cold S.W., and a three-way valve that bypassed the supply of hot F.W. to the HEX.

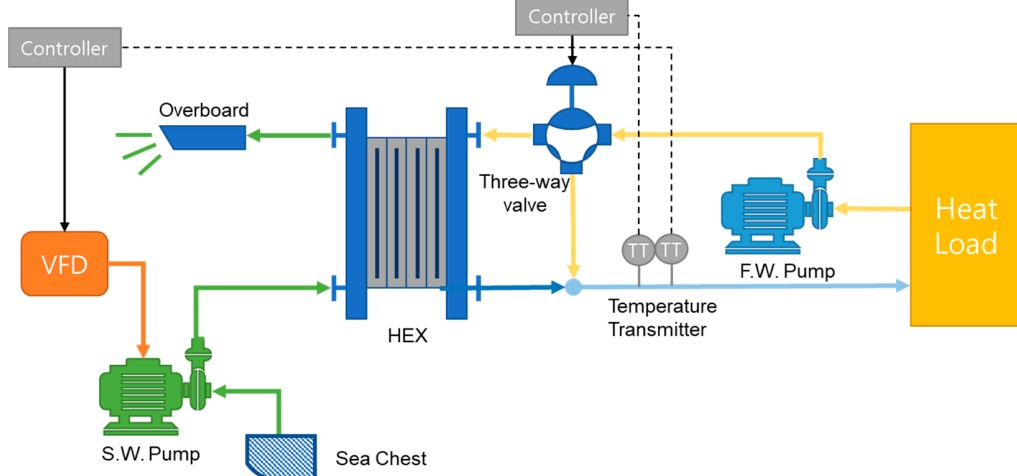

**Figure 1.** An example of a central cooling system on a ship.

The F.W. pump circulated the F.W. at a constant rotational speed to cool the heat load. The heated F.W., at approximately 38–43 °C, was supplied to the central cooling system. The three-way valve regulated the F.W. flow rate, determining whether it was supplied to the HEX or bypassed. The low-temperature S.W. cooled the F.W. supplied to the HEX. The cooled F.W. was mixed with the high-temperature F.W. that bypassed the HEX through the three-way valve. The combined F.W. then returned to the heat load.

On the other hand, the low-temperature S.W. was supplied to the HEX through the S.W. pump. The S.W. pump was driven by a variable-speed motor controlled by an inverter. By changing the voltage frequency of the input power supplied to the S.W. pump motor, the amount of S.W. supplied to the HEX could vary.

## 2.2. Configuration for Simulation

The central cooling system was implemented using Matlab R2020b Simulink [13–15], as shown in Figure 2.

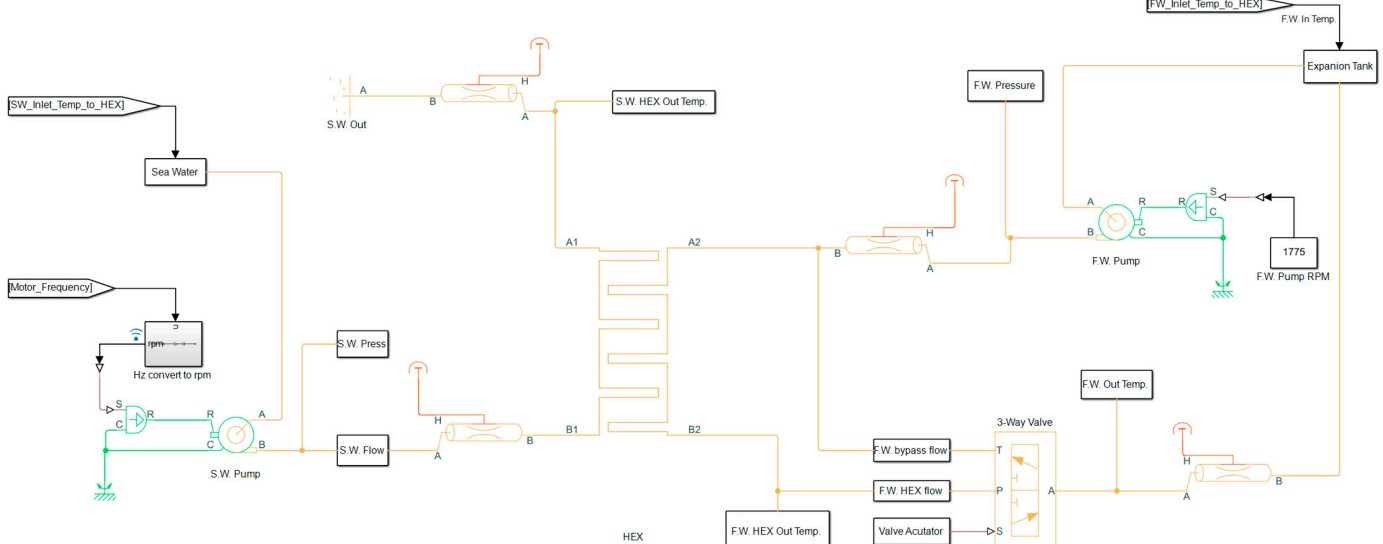

**Figure 2.** Modeling of the central cooling system.

The parameter values for each pump (F.W. and S.W. pumps), three-way valve, and HEX model were referred to in the datasheet information of each piece of equipment.

The F.W. pump was driven by a four-pole, three-phase, 440 V, and 60 Hz motor. Referring to the factory acceptance test (FAT) report and considering a slip, it was modeled to rotate at a constant speed of 1775 rpm. The S.W. pump also used a motor with the exact specifications of the F.W. pump. However, the frequency input to the S.W. pump motor varied from 30 to 60 Hz through an inverter. Considering the slip, the actual speed achievable by the S.W. pump motor was set to a maximum of 1775 rpm. The minimum speed was set at 887 rpm based on a motor input frequency of 30 Hz. Additionally, the suction pressures of the S.W. and F.W. pumps were set based on the pressure measured by the suction gauges when each pump was stopped.

The three-way valve directed the bypass flow of F.W. through the T port, F.W. from the HEX through the P port, and the combined F.W. to the output through the A port, as outlined in Figure 2. The bypass flow controlled by the actuator was set to linear variation. Furthermore, the offset of the T and P ports was set to 0 so that when the T port was fully open, the P port was completely closed, and vice versa.

The datasheet and the HEX FAT report referred to various necessary parameter values such as the HEX's surface area, overall heat transfer coefficient (O.H.T.C.), thermal conductivity, pressure drop, and pipe diameters.

### 2.3. Tuning for the Simulation Model

Based on Figure 2, the modeling was performed using the actual FAT reports and datasheets of each piece of equipment. However, it was challenging to obtain simulation results that closely resembled reality due to factors such as pipeline roughness, the thermal mass of the HEX, and wall thermal resistance, which were not available in the equipment's datasheet. Therefore, to achieve simulation results that approximated reality, additional parameter values not found in the datasheet were sought after using actual operating data.

The actual operating data included pressure data for the F.W. pump and temperature data for the F.W. outlet of the three-way valve, recorded at intervals of 10 s. The other data were collected at intervals of 60 s.

A simulation of the central cooling system using the actual operating data from Figure 3 was performed. The input data for the model included the voltage frequency of the S.W. pump motor, S.W. inlet temperature, opening degree of the three-way valve, and F.W. inlet temperature. From 0 to 1000 s, the opening degree of the three-way valve was fixed at 0.8, and only the rotational speed of the S.W. pump varied. Then, from 2000 to 2600 s, the rotational speed of the S.W. pump was fixed at its maximum, and only the three-way valve opening varied. Figure 3b represents the actual output when the central cooling system was inputted with the data from Figure 3a. MATLAB's Parameter Estimator was used to approximate the output of the modeled central cooling system to match Figure 3b.

### 2.4. Verification for the Simulation Model

The tuned simulation model was validated using another set of actual operating data. The input data included the voltage frequency of the S.W. pump motor, S.W. inlet temperature, opening degree of the three-way valve, and F.W. inlet temperature. The output data included the S.W. outlet temperature of the HEX, F.W. outlet temperature of the HEX, S.W. pressure, F.W. pressure, and F.W. outlet temperature of the three-way valve.

Among the results, the S.W. and F.W. outlet temperatures of the HEX and the F.W. outlet temperature of the three-way valve were compared with the corresponding simulated values obtained from the simulation.

Figure 4a shows the S.W. temperature at the HEX outlet, comparing the actual operating data with the simulation results. After the simulation started, in 300 s, the result was approximately 0.7 °C lower than the actual operating data.

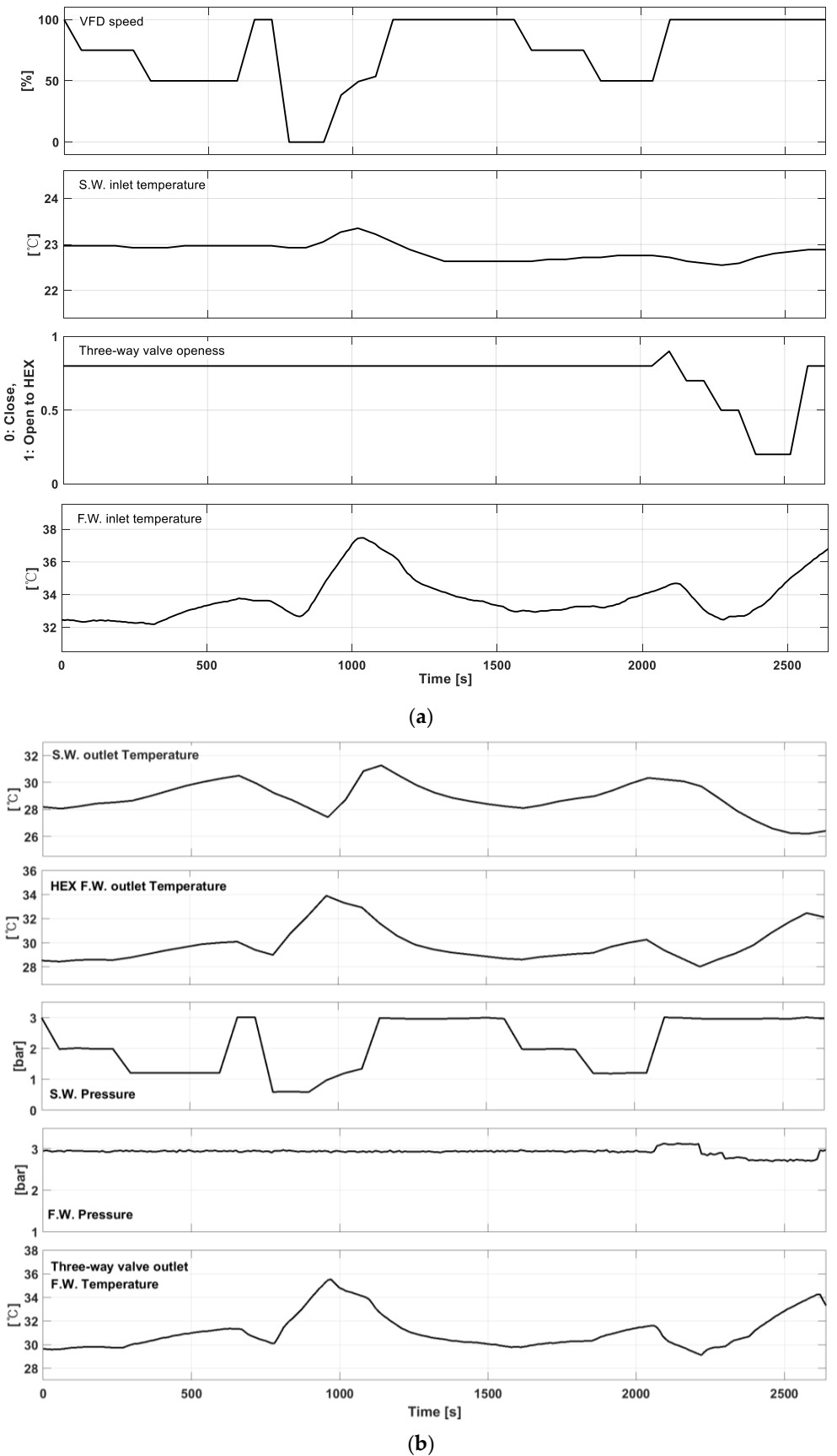

**Figure 3.** Operation data for the simulation model tuning of the central cooling system in a ship. (**a**) Operation input data for tuning; (**b**) operation output data for tuning.

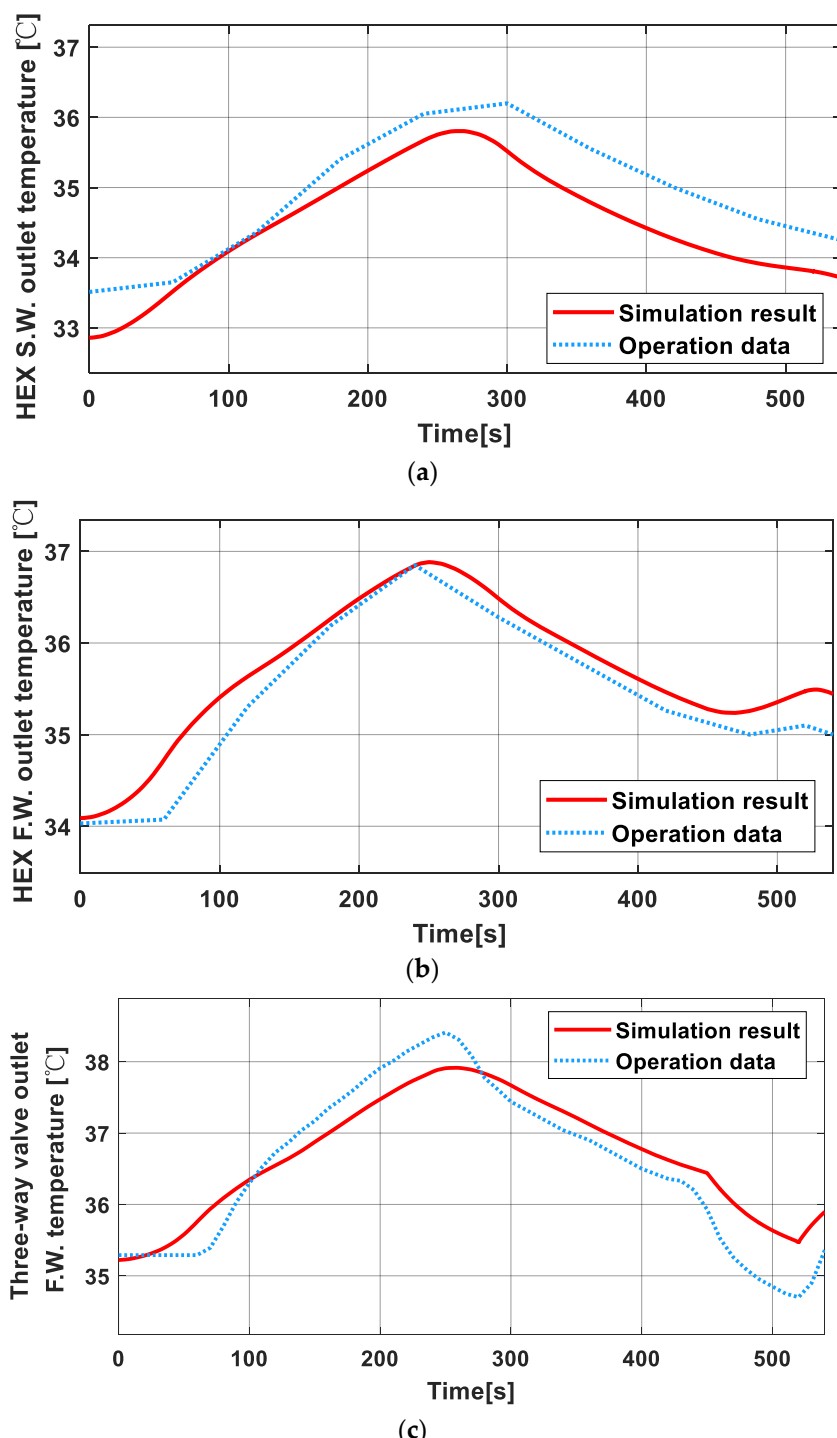

**Figure 4.** Comparing operation data and simulation output: (**a**) comparing HEX S.W. outlet temperature; (**b**) comparing HEX F.W. outlet temperature; (**c**) comparing controlled F.W. temperature at the three-way valve outlet.

Figure 4b shows the F.W. temperature at the HEX outlet, comparing the actual operating data with the simulation results. The maximum difference between the simulation results and operating data was +0.4 °C.

Figure 4c shows the F.W. temperature at the outlet of the three-way valve, comparing the actual operating data with the simulation results. The maximum deviation from the actual operating data was +0.7 °C, while the minimum deviation was −0.6 °C, using the actual operating data as a reference.

The fine-tuned simulation model verified through the simulation result was close to the operation data, as shown in Figure 4.

## 3. IMC-Based PI Controller Tuning Method using IMC Filters

The system transfer functions were required to design an IMC-based PI controller. In this case, we needed to determine the transfer functions of the system relating the voltage frequency input of the S.W. pump to the F.W. output temperature $G_{sw}(t)$ and the opening degree input of the three-way valve to the F.W. output temperature $G_{3v}(t)$. Even though the actual systems, $G_{sw}(t)$ and $G_{3v}(t)$, were not independent and could influence each other, it was supposed that they would not affect each other, and so we calculated their transfer functions accordingly. With this in mind, it was much easier to obtain the simple transfer functions. Figure 5 shows the transfer functions for $G_{sw}(t)$ and $G_{sw}(t)$.

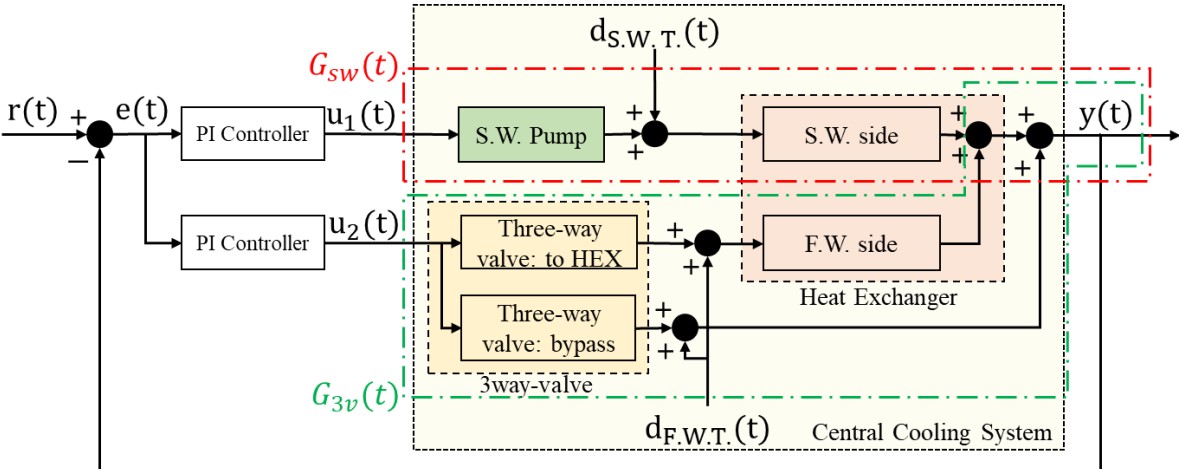

**Figure 5.** The control diagram of the central cooling system.

### 3.1. Finding Transfer Functions

A step input was applied for each of the two systems, and the transfer functions were estimated based on the results. First, to estimate $G_{sw}(t)$, the F.W. inlet temperature, S.W. inlet temperature, and three-way valve input were maintained at 39 °C, 22 °C, and fully opened as 1.0, respectively, while the input voltage frequency of the S.W. pump was stepped up from 30 to 40 Hz. The results are shown in Figure 6. Additionally, for $G_{3v}(t)$, the F.W. and S.W. inlet temperatures were kept the same at 39 °C and 22 °C, while the input power frequency of the S.W. pump was maintained at 60 Hz. The three-way valve openness was stepped up from 0.5 to 0.7. The resulting changes in the F.W. output temperature for the step input are also shown in Figure 6.

The transfer functions were obtained using MATLAB's Parameter Estimator based on the step responses. $G_{sw}(t)$ was estimated as a first-order system and $G_{3v}(t)$ was estimated as a second-order system, resulting in Equations (1) and (2), respectively. Furthermore, those expected transfer functions matched at a value of 99% in both systems. The abbreviations used for the IMC model are listed in Table 1.

$$G_{sw}(s) = \frac{y(s)}{u_1(s)} = \frac{-0.001709}{s + 0.01051} = \frac{-0.1627}{95.147s + 1} \tag{1}$$

$$G_{3v}(s) = \frac{y(s)}{u_2(s)} = \frac{-28.6s - 0.1456}{s^2 + 2.112s + 0.0165} = \frac{-8.871(197.05s + 1)}{(0.475s + 1)(128.205s + 1)} \tag{2}$$

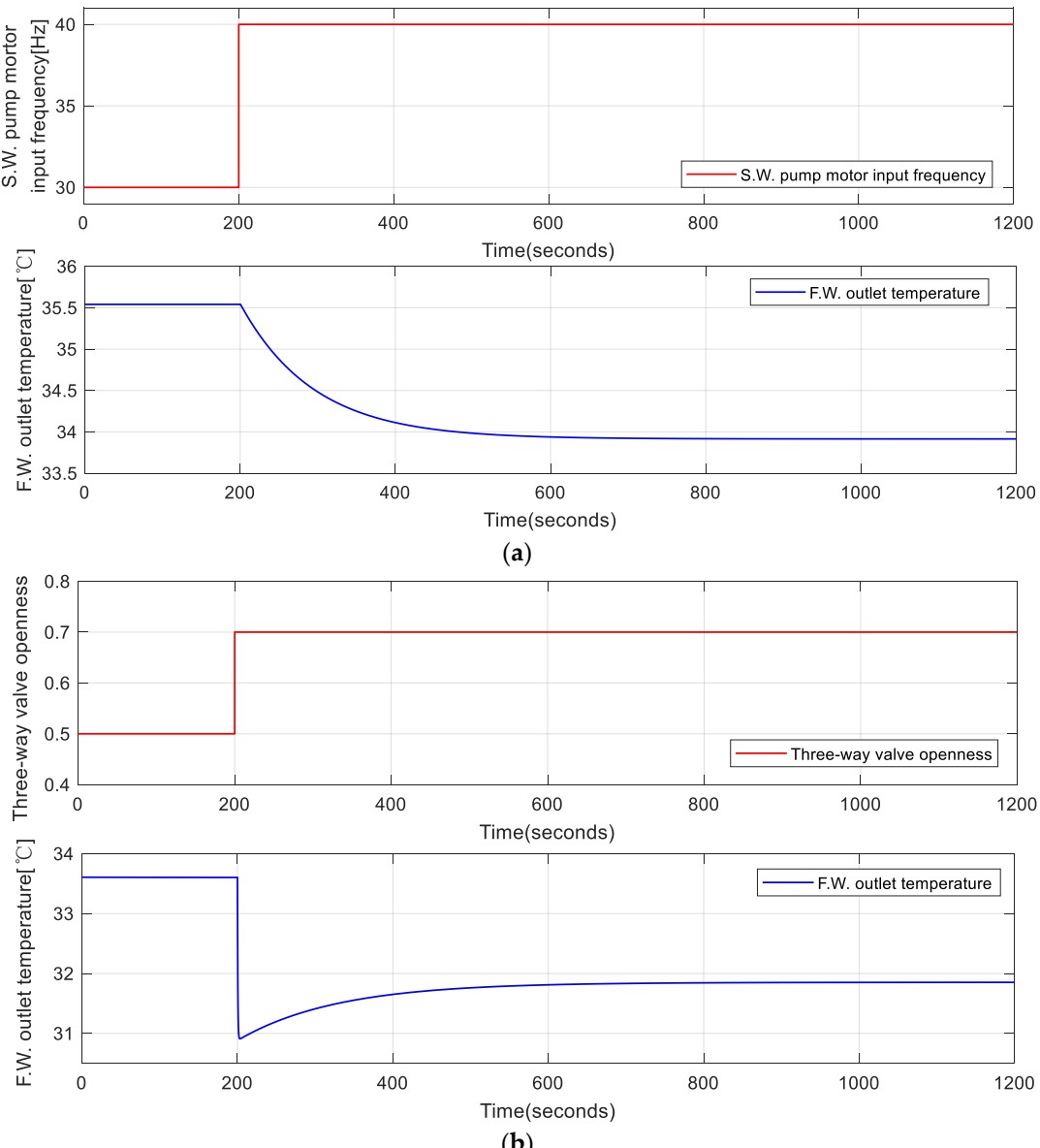

**Figure 6.** Step response of F.W. outlet temperature against S.W. pump speed and three-way valve openness. (**a**) Step response of $G_{SW}(t)$; (**b**) step response of $G_{3v}(t)$.

**Table 1.** Abbreviations.

| Symbol | Description |
|---|---|
| $f(s)$ | Filter function |
| $G(s)$ | Transfer function of a plant or a system |
| $k_{3v}, k_{sw}$ | Gain of transfer function |
| $K(s)$ | Control gain function |
| $K_i$ | Integral gain |
| $K_p$ | Proportional gain |
| $\lambda_{3v}, \lambda_{sw}$ | IMC filter |
| $q(s)$ | Invert plant and filter function |
| $\tau_b, \tau_p, \tau_s, \tau_{sw}$ | Time constant of plant transfer function |
| $T_i$ | Integral time |
| $u(s), u_1(s), u_2(s)$ | System input, S.W. pump motor voltage frequency input, and three-way valve openness input |
| $y(s)$ | System output |

### 3.2. Deriving Control Gains of IMC-Based PI Controllers

If $G_p(s)$ designates the plant to be controlled, the transfer function of the plant is denoted as $\widetilde{G}_p(s)$, and the product of the inverse of the plant transfer function and the filter is represented by $q(s)$, then the IMC principle can be represented as shown in Figure 7a. By considering $q(s)$ and $\widetilde{G}_p(s)$ as the control gain $K(s)$ within the dashed lines in Figure 7a, it can be represented as shown in Figure 7b. For the systems $G_{sw}(s)$ and $G_{3v}(s)$, their respective $K(s)$ are calculated to determine the $K_p$ and $K_i$ values of the PI controller [16].

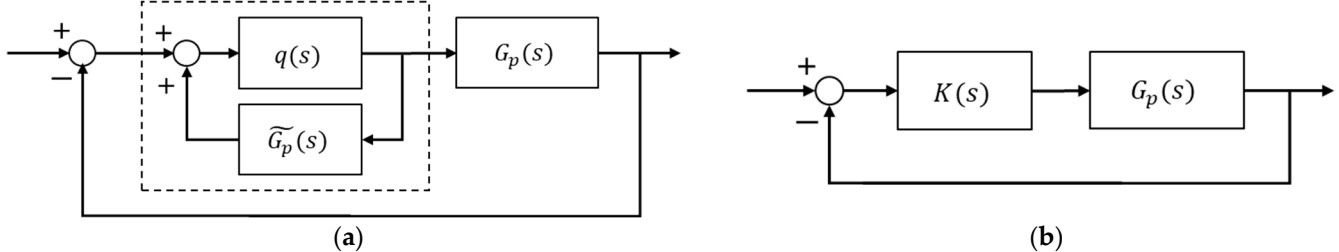

(**a**)　　　　　　　　　　　　　　　　　　(**b**)

**Figure 7.** Principle of IMC. (**a**) Principle of IMC model; (**b**) simplified IMC model.

#### 3.2.1. Sea Water Pump System

The general mathematical modeling of the first-order transfer function, the S.W. pump system $G_{sw}(s)$, is as follows, represented by Equation (3).

$$\widetilde{G_{sw}}(s) = \frac{k_{sw}}{\tau_{sw}s + 1} \tag{3}$$

Furthermore, the product of the inverse of the transfer function of the S.W. pump system and the filter, denoted as $q(s)$, is expressed as shown in Equation (4).

$$q(s) = \widetilde{G_{sw}}(s)^{-1}f(s) = \frac{\tau_{sw}s + 1}{k_{sw}} \frac{1}{\lambda_{sw}s + 1} \tag{4}$$

Therefore, the control gain $K(s)$ in Figure 7 can be written as shown in Equation (5).

$$K(s) = \frac{q(s)}{1 - \widetilde{G_{sw}}(s)q(s)} = \left(\frac{\tau_{sw}}{k_{sw}\lambda_{sw}}\right)\frac{\tau_{sw}s + 1}{\tau_{sw}s} \tag{5}$$

If the control gain $K(s)$ in Equation (5) is converted into the form of a PI controller, it becomes Equation (6).

$$K(s) = \frac{\tau_{sw}}{k_{sw}\lambda_{sw}} + \frac{1}{k_{sw}\lambda_{sw}}\frac{1}{s} = K_p + \frac{K_p}{T_i}\frac{1}{s} = K_p + K_i\frac{1}{s} \tag{6}$$

In that case, the gains $K_p$ and $K_i$ of the PI controller are given by Equation (7).

$$K_p = \frac{\tau_{sw}}{k_{sw}\lambda_{sw}}, \quad T_i = \tau_{sw} \ or \ K_i = \frac{1}{k_{sw}\lambda_{sw}} \tag{7}$$

#### 3.2.2. Three-Way Valve System

The transfer function of the second-order system, the three-way valve system $G_{3v}(s)$, is given by Equation (8):

$$\widetilde{G_{sw}}(s) = \frac{k_{3v}(\beta s + 1)}{(\tau_s s + 1)(\tau_b s + 1)} \tag{8}$$

where $\tau_s < \tau_b$.

Furthermore, the filter function is assumed to be represented by Equation (9).

$$f(s) = \frac{\gamma s + 1}{(\lambda_{3v} s + 1)^2} \tag{9}$$

Using the same approach as in Section 3.2.1 to derive $K(s)$ in the form of a PI controller, Equation (10) is obtained.

$$K(s) = \frac{q(s)}{1 - \widetilde{G}_{sw}(s)q(s)} = \frac{\frac{(\tau_s s + 1)(\tau_b s + 1)}{k_{3v}(\beta s + 1)} \frac{\gamma s + 1}{(\lambda_{3v} s + 1)^2}}{1 - \frac{k_{3v}(\beta s + 1)}{(\tau_s s + 1)(\tau_b s + 1)} \frac{(\tau_s s + 1)(\tau_b s + 1)}{k_{3v}(\beta s + 1)} \frac{(\gamma s + 1)}{(\lambda_{3v} s + 1)^2}}$$

$$K(s) = \frac{1}{k_{3v}\lambda_{3v}^2} \frac{(\tau_s s + 1)(\gamma s + 1)}{s\left(s + \frac{2\lambda_{3v} - \gamma}{\lambda_{3v}^2}\right)} \frac{(\tau_b s + 1)}{(\beta s + 1)} = \frac{1}{k_{3v}\lambda_{3v}^2\left(\frac{2\lambda_{3v} - \gamma}{\lambda_{3v}^2}\right)} \frac{(\tau_b s + 1)(\gamma s + 1)}{s\left(\left(\frac{\lambda_{3v}^2}{2\lambda_{3v} - \gamma}\right)s + 1\right)} \frac{(\tau_s s + 1)}{(\beta s + 1)} \tag{10}$$

If $\lambda_{3v} > 0$, $\gamma > 0$, and $\lambda_{3v} = \gamma$, it can be expressed as shown in Equation (11).

$$K(s) = \frac{1}{k_{3v}\lambda_{3v}} \frac{(\tau_b s + 1)}{s} \frac{(\tau_s s + 1)}{(\beta s + 1)} = \frac{\tau_b}{k_{3v}\lambda_{3v}}\left(1 + \frac{1}{\tau_s s}\right)\frac{(\tau_s s + 1)}{(\beta s + 1)} \tag{11}$$

If $\frac{(\tau_s s + 1)}{(\beta s + 1)} \frac{(\tau_s s + 1)}{(\beta s + 1)}$ represents a lead–lag filter, then,

$$K_p = \frac{\tau_b}{k_{3v}\lambda_{3v}}, \qquad T_i = \tau_b \ \text{or} \ K_i = \frac{1}{k_{3v}\lambda_{3v}} \tag{12}$$

### 3.3. Simulation Result of Applied IMC-Based PI Controllers

Based on the equations derived in Sections 3.2.1 and 3.2.2, each of the PI controller parameters $K_p$ and $K_i$ for the $G_{sw}(s)$ and $G_{3v}(s)$ systems were calculated with the IMC filters, as shown in Table 2. The values of $\lambda_{sw}$ and $\lambda_{3v}$ were determined through iteration until the F.W. outlet temperature was better controlled with the applied IMC filters than without them.

**Table 2.** The tuned parameters of IMC-based PI controllers.

|  | IMC Filter | $K_p$ | $K_i$ |
|---|---|---|---|
| PI controller for $G_{sw}(s)$ | $\lambda_{sw} = 0.85$ | $-688.003$ | $-7.2309$ |
| PI controller for $G_{3v}(s)$ | $\lambda_{3v} = 0.90$ | $-16.0579$ | $-0.1253$ |

The parameter values from Table 2 were applied to each controller, and the following conditions were used for the simulation:

- To test the performance of the controllers, step-up and step-down inputs were simultaneously applied for both F.W. and S.W. temperatures, representing disturbances.
- The F.W. inlet temperature was stepped up from 38 °C to 42 °C, referencing the operating range of the modeled ship, and maintained at 42 °C for 1500 s. Then, it was stepped down to 38 °C from 42 °C.
- The S.W. inlet temperature was stepped up from 20 °C to 25 °C, based on the S.W. temperature in the coastal areas of Korea during spring and autumn [17], and maintained at 25 °C for 1500 s. Subsequently, it was stepped down from 25 °C to 20 °C.
- The upper and lower limits of the S.W. pump motor voltage frequency for the rotational speed were set to their actual values, 60 and 30 Hz, respectively.
- The upper and lower limits for the three-way valve openness were set to their actual values, 1.0 (fully opened to the HEX side and fully closed to the bypass side) and 0 (fully closed to the HEX side and fully opened to the bypass side).

Usually, the tuning of the PI controller determines the combination values of P and I gains, which results in good output control performance. The advantage of applying the IMC filter is adjusting only one variable, which is the IMC filter. Then, this could determine

the gains of the PI controller with the desired control performance. When the IMC filter value was reduced, the system's response time decreased as per Equations (7) and (12). Therefore, by adjusting $\lambda_{sw}$ and $\lambda_{3v}$, the combinations of the four gains for the two PI controllers could be determined more easily than without the IMC filters.

Figure 8 compares the simulation results of applying IMC filters and those obtained without them. Both of the overall control performances were similar.

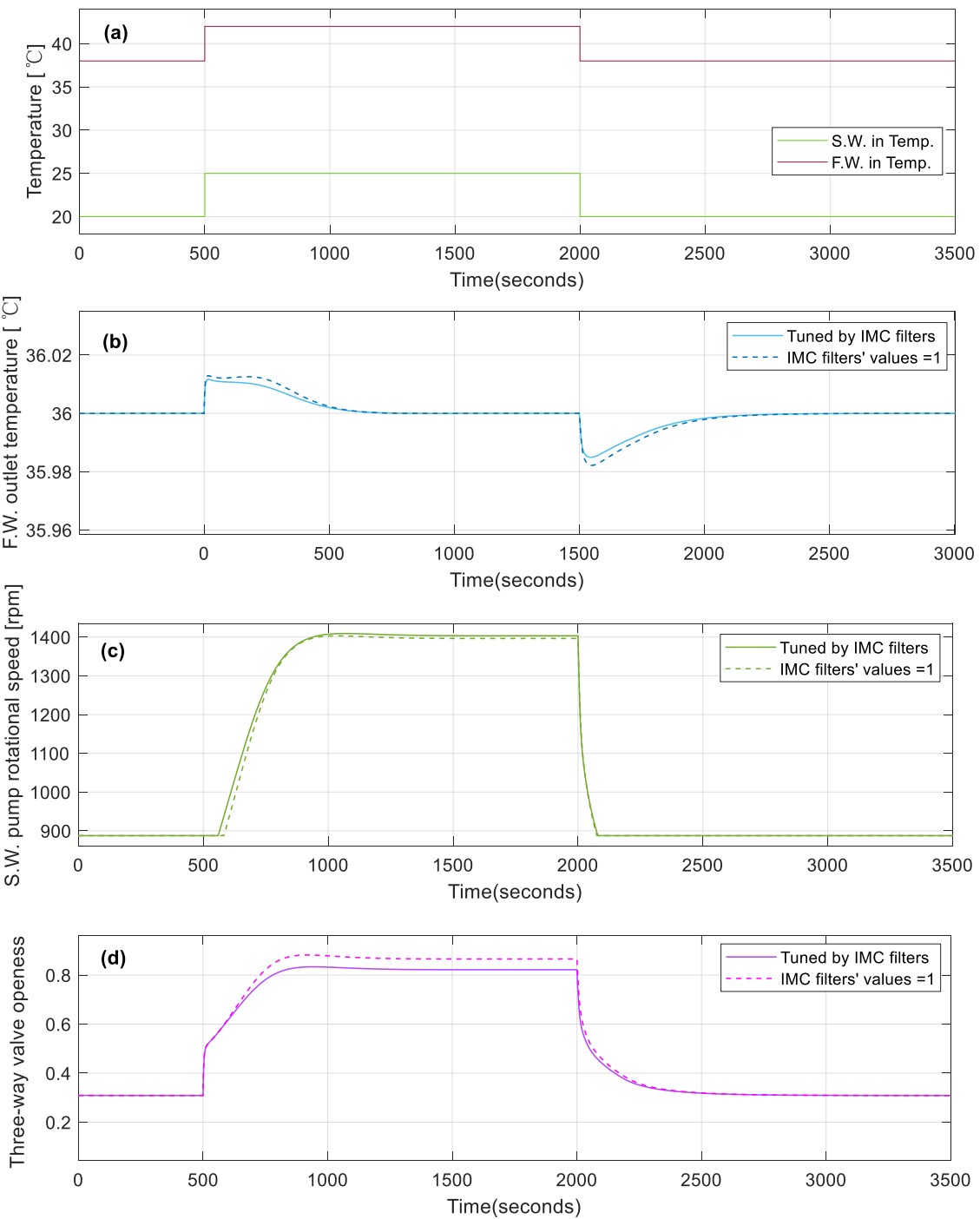

**Figure 8.** Simulation results of the applied IMC-based PI controllers. (**a**) F.W. and S.W. inlet temperatures, (**b**) F.W. outlet temperature, (**c**) S.W. pump rotational speed, and (**d**) three-way valve openness variation.

Figure 8a displays the variations in F.W. and S.W. inlet temperatures. At 500 s, the step-up inputs for both the F.W. and S.W. temperatures were applied. At 2000 s, the step-down inputs were applied.

Figure 8b shows the variation in the F.W. output temperature. Despite the step-up or -down disturbances, the F.W. output temperature of the tuned IMC filters remained within $\pm 0.02$ °C of the reference temperature, which was 36 °C. Additionally, after approximately 500 s following step disturbance, the F.W. output temperature recovered to the reference temperature. However, the F.W.-out temperature rose again after 200 s from the step-up disturbance applied in the case of the non-applied IMC filter. Supposedly, the reason for this was that the transfer functions modeling $G_{sw}(s)$ and $G_{3v}(s)$ were independent of each other.

Figure 8c displays the variations in the S.W. pump rotational speed. After step-up disturbance was applied, both rotational speeds increased after approximately 58 and 80 s. This delay occurred because it took around 58 and 80 s for the controller's output to exceed the minimum rotational speed of the S.W. pump. During step-up disturbance, the non-applied IMC filter showed a lower rotational speed than the tuned IMC filters.

Figure 8d shows the variations in the three-way valve openness. After step-up disturbance was applied, the three-way valves opened slightly more than 0.8. The tuned IMC filter showed less of an opening for the three-way valve than that for the non-applied IMC filter.

Reviewing Figure 8, the reason for the lower openness of the three-way valve and the higher S.W. pump rotational speed of the tuned IMC filter was because $\lambda_{sw}$ was set lower than $\lambda_{3v}$ to eliminate the re-rise of the F.W. output temperature.

## 4. Proposed Tuning Method with Modified IMC Guide

To utilize the IMC strategy for the tuning of the PI controller parameter, as discussed in Section 3.3, iterative simulations were required in order to find appropriate parameter values by adjusting $\lambda_{sw}$ and $\lambda_{3v}$ for the three-way valve openness controller and S.W. pump rotational speed controller. Additionally, by further reducing the rotational speed of the S.W. pump, which is related to achieving even more energy savings, it was necessary to fully open the three-way valve and minimize the bypass flow of fresh water after applying step disturbances.

Therefore, fine-tuning the parameters of the PI controller necessitated repeated simulations to determine the right balance between energy efficiency and effective F.W. temperature control performance, considering both the three-way valve control and S.W. pump speed control. Therefore, the following method was proposed for tuning the PI controllers.

Utilizing the characteristic that the F.W. output temperature response of $G_{sw}(s)$ against the frequency input variation in the S.W. pump motor was slower than the response of $G_{3v}(s)$ against the three-way valve openness input, the modified IMC strategy was suggested. Against a sudden disturbance, the F.W.-out temperature was controlled with the three-way valve's quick response. Then, the S.W. pump rotational speed was controlled with a slow change rather than the quick response of the rotational speed. Therefore, to eliminate the error of the desired F.W.-out temperature, the integral gain was determined to be more prominent than the proportional gain.

Initially, the proportional gain $K_p$ and integral gain $K_i$ of the PI controller were derived as per the standard IMC method. However, when deriving the PI controller gains of $G_{3v}(s)$, $\tau_s$ was used rather than $\tau_b$ in Equations (11) and (12). The IMC filter $\lambda_{3v}$ was set to one, which meant that the IMC filter was not used. And $\lambda_{sw}$ was set to the time constant of the S.W. pump transfer system $G_{sw}(s)$ in Equation (1). The suggested method was able to easily determine the control gains if the time constants of the transfer functions $G_{3v}(s)$ and $G_{sw}(s)$ were known compared with the standard IMC method outlined in Section 3.

Subsequently, for the PI controller related to the F.W. output temperature system concerning the S.W. pump power frequency input, the values of proportional and integral gains

were swapped, ensuring that the integral gain was more significant than the proportional gain per the control strategy. The detailed sequence of steps was as follows:

- For the PI controller of the F.W. output temperature system based on the three-way valve openness input, the gains were calculated using $\tau_s$, which was smaller than $\tau_b$ in Equations (11) and (12).
- The value of the IMC filter $\lambda_{3v}$ was set to one.
- The value of the IMC filter $\lambda_{sw}$ was set to 95, corresponding to the time constant value 95.147 from Equation (1), thereby determining the PI controller's integral gain and proportional gain values.
- The S.W. pump rotational speed controller's integral and proportional gains were exchanged with each other.

The above process ensured an effective tuning strategy, leveraging the distinct response characteristics of the system to the three-way valve openness and S.W. pump power frequency inputs. As a result, in response to abrupt changes in F.W. and S.W. input temperatures, the system rapidly restored the F.W. output temperature to the reference temperature by adjusting the openness of the three-way valve. Subsequently, any discrepancies in the F.W. output temperature caused by variations in the F.W. flow rate due to changes in the three-way valve openness were prevented from causing deviations from the reference temperature again. This was achieved through the significant integral gain of the controller, which led to adjustments in the S.W. pump rotational speed. By employing the method described above, the parameters of the PI controller were determined, as shown in Table 3.

**Table 3.** Parameters of proposed PI controllers.

|  | IMC Filter | $K_p$ | $K_i$ |
|---|---|---|---|
| PI controller for $G_{sw}(s)$ | $\lambda_{sw} = 95$ | −0.0647 * | −6.1558 * |
| PI controller for $G_{3v}(s)$ | $\lambda_{3v} = 1$ | −0.0535 | −0.1127 |

* Swapped $K_p$ and $K_i$ gains for PI controller of S.W. pump from the traditional IMC method.

*Simulation Result with Proposed PI Controllers*

The simulation conditions and methods remained consistent with those performed in Section 3.3. The subsequent Figure 9 compares the simulation results of the central cooling system using controllers tuned with the applied IMC filter discussed in Section 3.3 and the proposed method.

In Figure 9b, when subjected to step-up inputs for both F.W. and S.W. temperatures, the overshot of the F.W. output temperature using the proposed tuning method was approximately 0.3 °C higher than that of the IMC-based tuning method. Moreover, the proposed tuning method achieved a recovery time within 0.1 °C of the desired value for the F.W. output temperature in about 19 s. For step-down inputs for both temperatures, the proposed tuning method exhibited an overshot of about 0.1 °C more than the IMC-based tuning method, and the recovery of 0.1 °C was approximately achieved in 15 s. The applied IMC filter tuning method had a superior control performance for the F.W. output temperature against the disturbances in the F.W. and S.W. temperatures. In addition, if a smaller $\lambda_{3v}$ value were applied to the proposed tuning method with the modified IMC guide, it would be expected to increase the F.W.-out-temperature control performance.

However, the proposed tuning method operated the pump at a lower speed concerning the S.W. pump rotational speed, which was significantly related to efficient energy usage. As shown in Figure 9c, after a step-up disturbance input, comparing the S.W. pump rotational speed following the recovery of the F.W. output temperature, the proposed tuning method resulted in an approximately 9 rpm lower speed compared with the IMC-based tuning method. Before the step-up disturbance and after the step-down disturbance, the S.W. pump rotational speed remained at around 887 rpm for both methods. This was because the minimum voltage frequency of the S.W. pump motor inverter was 30 Hz. The larger

overshot in the F.W. output temperature using the proposed tuning method was due to the slower opening of the three-way valve compared with the IMC-based tuning method.

Furthermore, the three-way valve in the proposed tuning method, as illustrated in Figure 9d, was more open on the HEX side, regulating the flow to allow more F.W. water for cooling compared with the IMC-based tuning method.

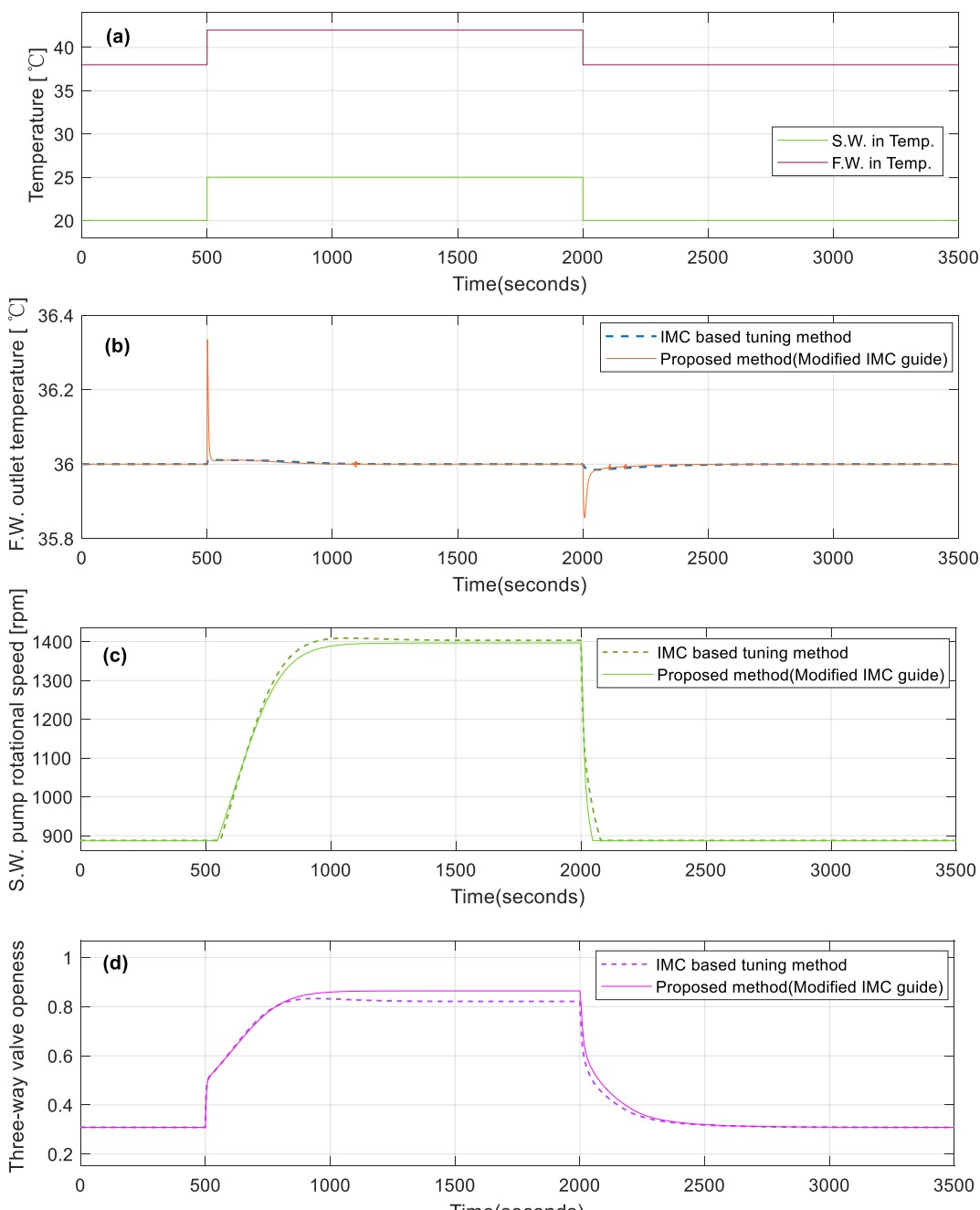

**Figure 9.** Comparing the simulation results of the proposed tuning method and the IMC-based tuning method. (**a**) F.W. and S.W. inlet temperature, (**b**) F.W. outlet temperature, (**c**) S.W. pump rotational speed, and (**d**) three-way valve openness variation.

## 5. Conclusions

The central cooling system used in ships was modeled for simulation tests. In order to improve the accuracy of the simulations, the central cooling system was tuned using actual operational data. The tuned central cooling system was validated using another

operational dataset. That process achieved a relatively realistic approximation of the central cooling system.

Transfer functions for the response of the F.W. output temperature from variations in S.W. pump rotational speed and three-way valve openness were derived for tuning PI controllers using the IMC guidelines. Despite the mutual influence between variations in the S.W. pump rotational speed and three-way valve openness on the F.W. output temperature, we approached simplifying this by supposing the independence of the two-control loops for IMC tuning. Appropriate controller gain values were obtained via multiple simulations using the derived transfer functions through the variation in the IMC filter values. Furthermore, the simulation test with step disturbances confirmed the efficient tuning configurations of the two PI controllers.

However, tuning the two controllers using IMC guidelines required finding the proper IMC filter values that allowed S.W. pump rotational speed and three-way valve openness to be efficiently controlled without interference. In order to improve a more open three-way valve and lower S.W. pump rotational speed, the modified IMC tuning method was proposed based on the characteristics of significant differences in the time constants between the transfer functions of the S.W. pump rotational speed control system and the three-way valve openness control system. Then, the test simulation confirmed the increased three-way valve openness and reduced the S.W. pump rotational speed against the step disturbances compared with the two PI controllers tuned using the IMC-based tuning method with the variation in the IMC filters. Further, the F.W.-out-temperature control performance was reasonable, and the way we determined the control gains was simple.

In conclusion, this study demonstrated two aspects regarding the application of two controllers in a central cooling system. Firstly, it presented a method to tune the gains of two PI controllers using the IMC approach, supposing that the two systems were independent. Secondly, it verified that the proposed tuning method for the two controllers enabled a more uncomplicated adjustment of the gains of the PI controller when the time constants were known and there was more efficient control of the S.W. pump rotational speed.

**Author Contributions:** Conceptualization, T.-Y.J.; data acquisition, B.-G.J.; methodology, T.-Y.J.; formal analysis, T.-Y.J.; writing—original draft preparation, T.-Y.J.; writing—review and editing, B.-G.J. All authors have read and agreed to the published version of the manuscript.

**Funding:** This research received no external funding.

**Institutional Review Board Statement:** Not applicable.

**Informed Consent Statement:** Not applicable.

**Data Availability Statement:** No applicable.

**Acknowledgments:** The central cooling system operation data are from the T/S Hanara Training Ship of the Korea Maritime and Ocean University.

**Conflicts of Interest:** The authors declare no conflict of interest.

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
