# Peer review of "A Study of PI Controller Tuning Methods Using the Internal Model Control Guide for a Ship Central Cooling System as a Multi-Input, Single-Output System"

_jmse, doi:10.3390/jmse11102025_

Round 1

Reviewer 1 Report

Dear Authors,

The presented study is engaging and has a novelty. The proposed method, of course, can find practical application in the operation of ships and provides an increase in the efficiency of ship cooling systems. The manuscript is generally executed very qualitatively and meets the journal’s requirements. The manuscript can be published in its present form.

Author Response

Dear Reviewer,

Thank you for your kind review of our study. Your feedback is appreciated, and we look forward to seeing our work in the journal.

Best regards,

Reviewer 2 Report

see attachment

Author Response

Dear reviewer,

Thanks for your valuable review.

The equation has been updated as you commented.

Best Regards,

Reviewer 3 Report

This paper demonstrates two aspects of the application of two controllers in a central cooling system, firstly, a method for tuning the gains of two PI controllers using the IMC method is proposed with the objective of achieving stable temperature control and reducing the motor speed of the pumps, and comparative simulation experiments are carried out to validate the effectiveness of the method. This is a meaningful work. But there are still the following problems:

1. There are some small grammar issues in the paper. Please check them carefully.

2, the IMC method presented in the paper can be compared with the results without the use of the method and does not demonstrate the superiority of the method.

3, there is a small problem with the insertion of references in the article, please revise it in time.

4, the role of coupling is not considered in the article, and how to verify the accuracy of the model is not specified in the article.

5, the labelling and axis annotations in the pictures can be as large as possible.

Author Response

This paper demonstrates two aspects of the application of two controllers in a central cooling system, firstly, a method for tuning the gains of two PI controllers using the IMC method is proposed with the objective of achieving stable temperature control and reducing the motor speed of the pumps, and comparative simulation experiments are carried out to validate the effectiveness of the method. This is a meaningful work. But there are still the following problems:

1. There are some small grammar issues in the paper. Please check them carefully.

Thanks for the comment. Reviewed and updated the manuscript.

2, the IMC method presented in the paper can be compared with the results without the use of the method and does not demonstrate the superiority of the method.

Dear Sir, the manuscript’s focus is two PI controller tuning methods. The first suggested IMC method. The second is a modified way from IMC guide. And it compares the first and second method-tuned PI controllers. The second tuned PI controller regulates the lower rotation speed of the S.W. pump against step disturbance. Further, the tuning method is easier than the first if the time constants are known. 

3, there is a small problem with the insertion of references in the article, please revise it in time.

Thanks for your comment. The reference style is updated, and the insertion of references are updated.

4, the role of coupling is not considered in the article, and how to verify the accuracy of the model is not specified in the article.

As stated in the manuscript, affecting between the two systems is ignored to develop easily the two transfer functions of the systems. The simulation model was tunned with the actual operation data set and verified the model with another operation data set as stated in 2.2 & 2.3 of the manuscript. Through the simulation model, the two transfer functions are achieved by step response and the estimator tool of Matlab. “those expected transfer functions match at 99% in both systems” is added.

5, the labelling and axis annotations in the pictures can be as large as possible.

As commented, the pictures, of which labeling and axis annotations are small, have been enlarged as recognizable.

We appreciate your valuable comments to improve the quality of the manuscript and hope the above response and the revisions are satisfactory to the reviewer's comments.

Best Regards,

Author Response

Dear Sir,

We appreciate your valuable comments and hope the answers will be enough to answer your questions.

Please note the attached file.

Best Regards,

Round 2

Author Response

Dear Sir,

Thanks for your valuable comments.

Please see the attached reply file.

Best Regards,

Round 3

Author Response

Dear Sir,

Thanks for your valuable comments, which made the paper has been upgraded.

Please see the reply to your comments and the updated manuscript.

Best Regards,
